# Impact of subjective well-being on physical frailty in middle-aged and elderly Japanese with high social isolation

Kai Tanabe[1,2]ᐤ*, Yuki Sugawara[3,4]ᐤ, Eiichi Sakurai[3], Yoichi Motomura[3], Yukihiko Okada[5,6], Akiko Tsukao[7], Shinya Kuno[1,2]

1 Faculty of Health and Sport Sciences, University of Tsukuba, Tsukuba, Japan, 2 R&D Center for Smart Wellness City Policies, University of Tsukuba, Tsukuba, Japan, 3 Artificial Intelligence Research Center, National Institute of Advanced Industrial Science and Technology, Tokyo, Japan, 4 Degree Programs in Systems and Information Engineering, Graduate School of Science and Technology, University of Tsukuba, Tsukuba, Japan, 5 Faculty of Systems and Information Engineering, University of Tsukuba, Tsukuba, Japan, 6 Center for Artificial Intelligence Research, University of Tsukuba, Tsukuba, Japan, 7 Tsukuba Wellness Research Co., Ltd., Chiba, Japan

ᐤ These authors contributed equally to this work.
* tanabe.kai.gm@u.tsukuba.ac.jp

**Data Availability Statement:** Data cannot be shared publicly because of the contract with the Ministry of Health, Labor and Welfare (MHLW). Data are available from the MHLW Grants system

## Abstract

Social isolation exacerbates physical frailty and is associated with subjective well-being. Even those with high levels of social isolation may have different health statuses depending on the type of isolation and their subjective well-being. However, the effect of subjective well-being on the relationship between social isolation and physical frailty remains unclear. This study examined whether the risk of physical frailty was the same for individuals with social isolation according to high and low subjective well-being. The study participants included 1,953 middle-aged Japanese adults aged 45 years and older. Physical frailty was assessed using a modified version of the Fried phenotype criteria. Probabilistic Latent Semantic Analysis was used to classify participants according to social isolation indicators. Subsequently, we focused on the groups with high social isolation and classified them according to whether their subjective well-being was high or low. Subjective well-being was evaluated using the *Shiawase* and *Ikigai* scales, which are concepts used in Japan. Finally, we used survival time analysis to examine the relationship between *Shiawase* or *Ikigai* and physical frailty in groups with high social isolation. The participants were classified into four groups based on their social isolation status. The physical frailty rate of the high social isolation class was 37.0%, which was significantly higher than that of the other classes. Survival time analysis revealed that among people with high social isolation, those with high *Shiawase* and *Ikigai* had a significantly lower risk of physical frailty than those with low *Shiawase* and *Ikigai*. All individuals with high social isolation are not at a high risk of physical frailty. The findings reveal that even those with high level of social isolation may have a lower risk of physical frailty if their subjective well-being is high. These results will contribute to promoting the prevention of frailty in middle-aged and older adults.

(contact via e-mail; mhlw-grants@niph.go.jp) for researchers who meet the criteria for access to confidential data. The data underlying the results presented in the study are available from MHLW Grants system. The contact information for the MHLW Grants system is listed below. MHLW Grants system 2-3-6, Minami, Wako, Saitama 351-0197, Japan e-mail: mhlw-grants@niph.go.jp URL: https://mhlw-grants.niph.go.jp/ Tel: +81-4-8458-6210

**Funding:** This work was supported by a Health Labor Sciences Research Grant, Research for Building Evidence on the Results of Verification Programs for Prevention and Health Promotion (22FB1002).The funders had no role in study design, data collection and analysis, decision to publish, or preparation of the manuscript.

**Competing interests:** The authors have declared that no competing interests exist.

## Introduction

Aging of the global population is the most important medical and sociodemographic issue that needs to be addressed not only in developed regions where the population is already aging but also in developing regions where the population is aging soon [1]. Japan has the highest percentage of population aged 65 years and older and has been called the advanced aging country; the country's actions and future are attracting global attention. In Japan, the number of households headed by elderly people aged 65 years and older accounts for 49.4% of all households, of which 28.8% are one-person households; this percentage is increasing every year [2]. Population decline and aging in rural and mountainous areas precede those in urban areas, and rural and mountainous areas face a lack of resources to support the social isolation of the elderly. Physical frailty is a health outcome that should be considered in the context of social isolation in older adults. Physical frailty, the leading complaint of frailty, has been reported to increase the risk of mortality and shorten healthy life expectancy, and has been shown to be associated with other health outcomes such as gait dysfunction, falls, fractures, cognitive decline, hospitalization, nursing home placement, and quality of life [3, 4]. However, physical frailty can be prevented because it is reversible [3]. Prevention of physical frailty not only contributes to extending healthy life expectancy and improving the quality of life of individuals but also helps solve social issues by reducing social security costs, such as medical and long-term care costs [5].

Several studies have examined the relationship between physical frailty and social isolation in older adults. Recently, Kojima et al. (2022) [6] reported the results of a systematic review and meta-analysis that provided evidence that social isolation contributes to an increased risk of physical frailty in older adults. In addition, a 14-year longitudinal study of a British population showed that social isolation was associated with an increased risk of developing physical frailty [7].

Social Isolation is a diverse concept [8, 9]. Recently, attention has been paid to this diversity when considering support for social isolation. Previous studies have typologies for diverse social isolation and have attempted to identify groups at higher risk of declining mental and physical health status [10, 11]. Barnes et al. (2022) [12] examined the impact of the combination of social isolation and loneliness on wellbeing. They reported lower well-being in the order of isolation only, loneliness only, and both compared to groups with neither social isolation nor loneliness. Smith and Victor (2019) [13] used latent class analysis to classify study participants based on three factors: social isolation, loneliness, and living alone. The results showed that the participants could be categorized into six clusters, of which the group that experienced social isolation and loneliness had poorer physical and mental health. Similarly for physical frailty, there is a need to consider the diversity of social isolation in the prevention of physical frailty. However, only a few studies have clarified this issue [14].

When examining the diversity of social isolation among the elderly in Japan, it is necessary to consider Japan-specific subjective well-being. Kumano (2018) [15] suggested subjective well-being in Japanese culture includes not only *Shiawase* (Happiness) as a Hedonic well-being, but also *Ikigai* (the joy and goal of living / a life worth living) as an Eudaimonic well-being. Several studies support this claim [16–18]. *Shiawase*'s emotions contain strong elements of joy and comfort and are oriented toward the present. On the other hand, the feeling of *Ikigai* has a strong element of devotion to one's favorite activities and is oriented toward the future. Previous studies have reported that *Shiawase* and *Ikigai* are associated with social isolation, but only a few studies have examined their relationship to physical frailty [19–21].

Therefore, the research question in this study was as follows: Does subjective well-being affect physical frailty in middle-aged and elderly Japanese with high social isolation? This study examined the impact of subjective well-being on physical frailty in middle-aged and elderly Japanese individuals living in mountainous areas with high social isolation.

## Methods

### Participants

This study was carried out in part by a Health Labour Sciences Research Program, Research for Building Evidence on the Results of Verification Programs for Prevention and Health Promotion. A portion of the cross-sectional study data was used. The data obtained for this project were a combination of survey data and data from the medical and long-term care insurance receipts of survey respondents [22]. The participants were middle-aged older adults aged 45 years or older living in a small Japanese city (population of approximately 40, 000 people) located in a mountainous area. These participants were also enrolled in either the National Health Insurance or Late-Stage Medical Insurance for the Elderly.

Participant recruitment and questionnaire surveys were conducted between October 15 and November 19, 2021. Medical and long-term care receipt data were collected on December 24, 2021, and combined with questionnaire data. The survey was conducted in a way that all citizens aged 45 years and older were surveyed. Questionnaires were sent to 11,303 individuals and data were collected from 4,045 participants. The data that could be combined with the respondents' medical and long-term care insurance receipts were 3,706. The total number of participants in the analysis was 1,953. Exclusion criteria for the analysis were missing questionnaire items assessing physical frailty and social isolation and having been certified as needing long-term care between 2011 and 2019. To assess the risk of physical frailty, individuals certified as requiring nursing care were excluded from the analysis. In addition, to eliminate the influence of COVID-19 as much as possible, participants were asked to recall two time points: 2019 (before the spread of COVID-19) and 2011. The variables recalled were frequency of conversation, participation in social activities, status of life activities, physical fitness, social capital in the community, and subjective well-being (*Shiawase* (happiness) and *Ikigai* (the joy and goal of living/a life worth living)). The questionnaire data included demographic variables, frequency of conversations, participation in social activities, status of life activities, physical fitness, social capital in the community, and subjective well-being. The medical and long-term care insurance receipt data included the date of birth, sex, and whether the patient was certified as needing long-term care.

### Measures

**Physical frailty.**   In this study, we assessed physical frailty using Fried's criteria [23]. Frailty was defined as the meeting of three or more of the following five evaluation criteria: shrinkage, exhaustion, low activity, slowness, or weakness. The frailty assessment index has changed frequently in previous studies, and it has been suggested that all details regarding the measurement of the frailty phenotype criteria should be reported to aid in the interpretation of results [24]. Therefore, in this study, we modified and applied Fried's assessment criteria (Table 1). Questionnaire data were collected in 2021 as a reminder of the 2019 conditions, and data from 2019 were used to assess physical frailty.

**Social isolation.**   Social isolation is an objective concept that refers to a state of limited social connection or assistance [25]. Previous studies have shown that social isolation is associated with increased mortality, cardiovascular diseases, and reduced functional status [26–28]. Questions have been used to measure social isolation, such as marital status; household composition; frequency of contact with friends, family, and children; and participation in social activities [7, 27, 29]. However, there are no uniform indicators of social isolation, because the items used to define social isolation vary across studies [30]. Therefore, in this study, we assessed social isolation using 22 questions, including job availability, frequency of phone and

**Table 1. Frailty assessment criteria.**

| Evaluation criteria | Fried et. al. (2001) [23] | Modified |
|---|---|---|
| Shrinkage | Baseline: >10 lbs lost unintentionally in prior year | Weight loss of at least 5% from 2011 to 2019 |
| Exhaustion | Self-reported exhaustion, identified by two questions from the CES–D scale | The total score of 21 points or more on a 6-point scale for "I felt cheerful and happy," "I felt calm and relaxed," "I was motivated and active," "I rested well and woke up feeling good," and "There were many things in my daily life that interested me." |
| Low activity | Kcals/week: lowest 20% males: <383 Kcals/week females: <270 Kcals/week | No to both "Did you engage in light sweaty exercise for at least 30 minutes a day for at least 2 days a week for at least 1 year?" and "Did you engage in walking or equivalent physical activity for at least 1 hour a day in your daily life?" |
| Slowness | Walking time/15 feet: slowest 20% (by gender, height) | No to "Do you think you walk faster than your peers of approximately the same age?" |
| Weakness | Grip strength: lowest 20% (by gender, body mass index) | No to one of the following questions: "Are you able to put on socks, pants, or a skirt while standing without support?" or "Are you able to lift a 10 kg bag of rice?" |

SNS conversations, frequency of going out less frequently, willingness to contribute to the community, and community cooperation trust, in addition to questions measuring social isolation used in previous studies. The questions assessing social isolation are listed in Table 2.

**Table 2. Social isolation assessment criteria.**

| Question type | Variables | Discretization | Score |
|---|---|---|---|
| Demographics | Living alone | yes or no | yes: 1, no: 0 |
| | Marital status | having a spouse, bereavement, divorce, unmarried, others | having a spouse: 1, the others: 0 |
| Frequency of conversation | Frequency of conversation with family | everyday or not | everyday: 0 not everyday: 1 |
| | Frequency of conversation with non-family | three point scale | high: 1, middle: 2, low: 3 |
| | Frequency of conversation with phone or SNS | | |
| Social activity | Participation of social activity | yes or no | yes: 0 no: 1 |
| | Participation of sports activity | yes or no | |
| | Participation of health and medical volunteer | yes or no | |
| | Participation of volunteer at elementary and junior high schools | yes or no | |
| | Participation of other volunteer | yes or no | |
| | Participation of hobby activity | yes or no | |
| | Participation of neighborhood association, residents' associations etc. | yes or no | |
| | Participation of learning and culture activity | yes or no | |
| | Participation of local event | yes or no | |
| | Participation of activity to convey special skills to others | yes or no | yes: 0 no: 1 |
| | Participation of other activity | yes or no | |
| Living conditions | Eating with someone once a day | yes or no | yes: 0 no: 1 |
| | Frequency of going out decreasing | yes or no | yes: 1 no: 0 |
| | Employed | yes or no | yes: 0 no: 1 |
| Community cooperating trust | Community cooperating trust | three point scale | high: 1, middle: 2, low: 3 |
| Willingness of contribution to the community | Willingness of contribution to the community | three point scale | |

We measured the frequency of conversation on a 5-point scale from "daily" to "rarely/never" in response to the question, "How often did you talk on the phone or social networking sites?" It was discretized into three levels: "every day, 5–6 days a week," "3–4 days a week," and "1–2 days a week, hardly ever/never." We measured willingness to contribute to the community on a 5-point scale from "applicable" to "not applicable" to the question "I have always wanted to contribute to the community." It was discretized into three levels: "Yes/somewhat," "Neither agree nor disagree," and "Not really/not really." We measured community cooperating trust on a 5-point scale from "very much so" to "not at all" in response to the questions "There are people in this community that I can cooperate, consult, and rely on" and "I think people in this community are generally trustworthy." We defined a total score of 9 or higher as high in community cooperating trust, 4–8 as normal in community cooperating trust, and 3 or lower as low in community cooperating trust.

**Subjective well-being.** Although the concept of well-being has been widely discussed, its meaning has not been defined as it varies across cultures. Kumano et al. (2018) [15] argue that in Japan, subjective well-being is expressed as both *Shiawase*, which is Hedonic well-being, and *Ikigai*, which is Eudaimonic well-being. The former is translated into English as "happiness" and the latter as "the joy and goal of living" or "a life worth living." These two well-being dimensions have been reported to be associated with health outcomes among Japanese people and investigated through national surveys [31–33].

In this study, *Shiawase* and *Ikigai* were evaluated based on the previous survey scales [15, 31, 34]. *Shiawase* was evaluated based on the question, "To what extent did you feel *Shiawase*?" It uses an 11-point scale ranging from 0 to 10, with higher scores indicating greater happiness. Scale scores of 0 to 4 were classified as low *Shiawase*, 5 to 6 as normal *Shiawase*, and 7 to 10 as high *Shiawase*.

*Ikigai* was evaluated based on the question, "To what extent did you feel *Ikigai*?" This scale was measured on a 5-point scale from "I feel well" to "I don't feel well at all." Scale scores of 1 to 2 were classified as high *Ikigai*, 3 as normal *Ikigai*, and 4 to 5 as low *Ikigai*.

## Analysis flow

We used probabilistic latent semantic analysis (PLSA) and survival time analysis to examine the associations between physical frailty, social isolation, and subjective well-being. First, we extracted potential clusters using the PLSA with questions on social isolation, as listed in Table 2. PLSA was proposed by Hofmann as a document classification method [35]. This method assumes that words $w_j$ in sentence $d_i$ are generated by the latent variable $z_k$. Likelihood maximization with the EM algorithm, with the latent variable $z \in Z = \{z_1,\ldots,z_k\}$ to accompany the co-occurring data. The simultaneous probability of sentence $d_i$ and word $w_j$ is expressed in Eq (1) using the latent variable $z_k$.

$$P(d_i, w_j) = \sum_k P(d_i|z_k)P(w_j|z_k)P(z_k) \tag{1}$$

The EM algorithm is then used to calculate $P(d|z)P(w|z)P(z)$, which maximizes the following log-likelihood function L as shown in Eq (2). In addition, $n(i,j)$ is the number of times word $w_j$ co-occurs with document $d_i$.

$$L = \sum_i \sum_j n(i,j) log P(d_i, w_j) \tag{2}$$

There is an initial value dependence because the likelihood calculation for PLSA uses the EM algorithm. Therefore, the number of iterations was set to 1,000, the initial value was given

three times, and the number of latent classes was increased from 1 to 10 to search for the best model. The number of latent classes was determined based on the Akaike's Information Criterion (AIC). The features of each cluster obtained by the PLSA were analyzed by constructing a Bayesian network model and applying a probabilistic inference algorithm based on loopy belief propagation. In this model, we used probabilistic inference to determine the direction of the contribution of question choices based on the positivity or negativity of the probability of belonging, and identified the characteristics of each cluster. An analytical method combining PLSA and Bayesian networks was developed by the National Institute of Advanced Industrial Science and Technology (AIST) in Japan and has been used in many empirical studies [36]. Ide et al. (2017) [37] used a questionnaire and ZIP code data of elderly people in 24 municipalities in Japan and extracted potential regional characteristics as segments. Furthermore, a Bayesian network was constructed using the questionnaire data and variables of the obtained segments, and probabilistic inferences were made. Consequently, this study reveals the potential regional characteristics and factors contributing to regional disparities. Kawai et al. (2022) [22] applied PLSA to questionnaire data, such as those used in this study, and classified people according to their health literacy. They also used a Bayesian network to construct a prediction model for physical frailty in each health literacy category. Consequently, this study enabled the construction of models with high predictive accuracy. Extracting potentially common meaningful classes using PLSA allows useful model building in subsequent analyses. In this study, we applied this method to 1,953 questionnaires and classified people into several clusters according to their social isolation, with sentence $d_i$ as the respondent of the questionnaire and word $w_j$ as the state of whether they selected each questionnaire item. We then analyzed the characteristics of each cluster. We analyzed the differences in sex, age group proportions, and physical frailty rates between clusters obtained by PLSA using the Z-test, with p-value adjustment using the Bonferroni method. We also analyzed differences in mean age between clusters using a one-way analysis of variance (ANOVA), as normality was confirmed using the Shapiro-Wilk test.

Next, we focused on clusters with high social isolation among the obtained clusters and analyzed the association between subjective well-being and physical frailty using survival time analysis. Survival time analysis is an analytical method that focuses on the relationship between events and the time until a specific event occurs, such as time to death, disease onset, and time to recovery. In this study, we used the Kaplan-Meier method, which represents the evolution of event rates with respect to the observation period, among survival time analysis methods. The observation period was defined based on the age of each individual and the occurrence of an event was defined as physical frailty. The log-rank test was used to determine statistical significance. *Shiawase* and *Ikigai* were used to assess subjective well-being. In the analysis of *Shiawase*, a total of 1,950 data were used, excluding three cases in which questions assessing *Shiawase* were missing. In the analysis of *Ikigai*, a total of 1,953 data were used because there were no missing questions that assessed *Ikigai*. We used PLASMA and Bayonet, an intellectual property software from the National Institute of Advanced Industrial Science and Technology, to run the PLSA and Bayesian network [38, 39]. We used IBM SPSS Statistics (Version 28.0.1.0) for the other data analysis. Statistical significance was set at p < 0.05.

## Results

### Comparing the physical frailty rates by social isolation cluster

Of the 1,953 participants included in the analysis, 996 (51.0%) were male and the mean age was 69.8 years. The averages and standard deviations for age, *Shiawase* score, *Ikigai* score, physical frailty score, and social isolation score are shown in Table 3. Social isolation scores in this study were calculated by summing the scores of the social isolation questionnaire items

**Table 3. Descriptive statistics of the analyzed data.**

| Question types | | n | Average (standard deviation) |
|---|---|---|---|
| Age (years) | | 1,953 | 69.8 (8.4) |
| *Shiawase* score | | 1,936 | 7.35 (2.23) |
| *Ikigai* score | | 1,942 | 1.95 (0.98) |
| Physical frailty score | | 1,953 | 1.44 (1.20) |
| Social isolation score | | 1,953 | 18.85 (3.32) |
| Question types | Variables | n | rate |
| Employment (n = 1,953) | Employed | 679 | 34.8% |
| Education status (n = 1,924) | Elementary and junior high school graduates | 421 | 21.9% |
| | High school graduate | 1,092 | 56.8% |
| | Specialized or vocational school graduate | 164 | 8.5% |
| | Junior college or technical college graduate | 87 | 4.5% |
| | University graduate | 153 | 8.0% |
| | Master's graduate | 7 | 0.4% |

(Table 2), in which 1 point was allocated for each of the following: living alone; not having a spouse; frequency of conversation with family members is not daily; not participating in any of the social activities, respectively; not eating with someone once a day; frequency of going out decreasing; not employed. In addition, the frequency of conversation with non-family and the frequency of conversation with phone or SNS, community cooperating trust, willing of contribution to community was divided into three-point scales: high, normal, and low, and assigned a score of 1, 2, and 3, respectively. In addition, the number and percentage of people in the analyzed data for employment status and education status are shown in Table 3.

The results of the PLSA using questions on social isolation showed that the AIC score was the lowest when K = 4. We adopted four classes 4, C1–C4. The details of the AIC scores are provided in S1 Fig. The number of persons belonging to each cluster, as well as their gender, age category, and mean age are shown in Table 4. C1 included 155 participants (63.9% male), C2 701 participants (52.6% male), C3 851 participants (47.4% male), and C4 246 participants (50.8% male). Sex and age group percentages are significantly different at the 5% level between groups with different symbols. The mean age for all age groups was 69.5 years for C1, 68.6 years for C2, 70.6 years for C3, and 70.4 years for C4. One-way analysis of variance confirmed a significant difference in mean age among the four groups C1–C4 (p<0.01).

Next, we constructed a Bayesian network model using the variables of the four clusters and the social isolation question to confirm the characteristics of the four clusters obtained using the PLSA. Fig 1 shows the nodes around C1 cluster in the constructed Bayesian network. The

**Table 4. Percentage of sex and age group and average age by cluster.**

| cluster | n | Percentage of sex (Male)[#] | Percentage of age [#] | | | Average age (years) (standard deviation) |
|---|---|---|---|---|---|---|
| | | | under 65 | over 65, under 75 | over 75 | all |
| C1 | 155 | 63.9%[a] | 19.3%[a,b,c] | 52.9%[a] | 27.7%[a,b] | 69.5 (8.34) |
| C2 | 701 | 52.6%[a,b] | 26.0%[c] | 50.4%[a] | 23.7%[b] | 68.6 (8.52) |
| C3 | 851 | 47.4%[b] | 15.9%[b] | 54.4%[a] | 29.7%[a] | 70.6 (7.50) |
| C4 | 246 | 50.8%[a,b] | 25.6%[a,c] | 38.2%[b] | 36.2%[a] | 70.4 (10.57) |
| all | 1,953 | 51.0% | 21.0% | 50.8% | 28.2% | 69.8 (8.42) |

[#] Groups with different symbols (a-c) indicate significant differences at the 5% level.

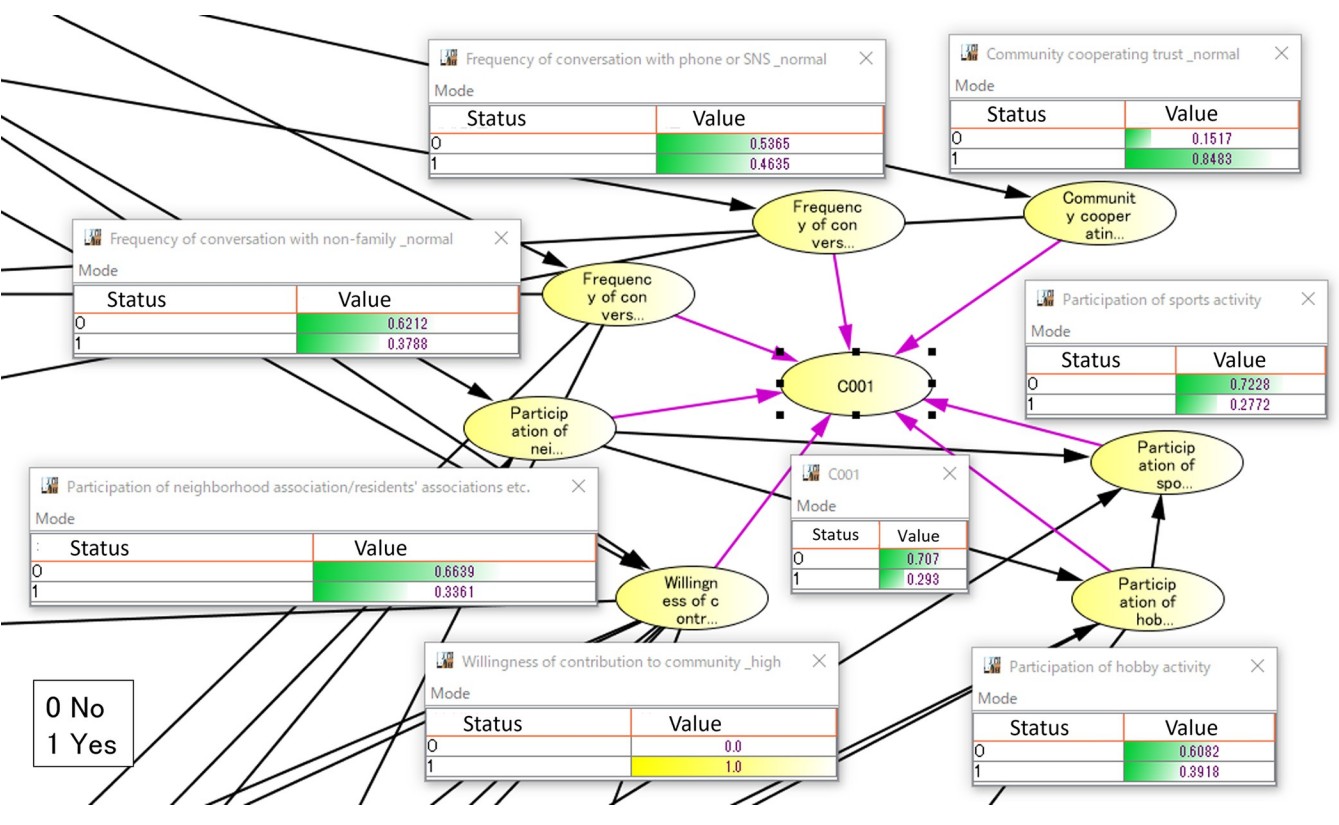

**Fig 1. Bayesian network about C1.**

nodes around C2–C4 are shown in S2–S4 Figs. The nodes with arrows pointing directly to each cluster are choices for questions in which the probability of belonging to each cluster changes. In this model, we used probabilistic inference to determine the direction of the contribution of question choices based on the positivity or negativity of the probability of belonging, and identified the characteristics of each cluster. The results of the probabilistic inference of the relevant questions for C1–C4 are shown in S1 Table. We used this probability inference to determine that C1 is characterized by "participates in sports and hobby-related activities" and "has a high sense of community contribution," C2 is characterized by "frequent conversations outside the family and on the phone and SNS" and "does not have a high sense of community contribution," C3 is characterized by "normal sense of community contribution" and "does not participate in social activities," and C4 is characterized by "does not participate in social activities" and "infrequent conversations outside the family, on the phone, and on SNS" and "low awareness of community contribution." These differences in characteristics suggest that C1 is a cluster with low social isolation and C4 is a cluster with high social isolation.

Table 5 shows the results of the comparison of physical frailty rates and the number of people with physical frailty for each of the four clusters. We observed that C4, with high social isolation, had a significantly higher rate of physical frailty than the other three clusters did.

### Comparing the risk of physical frailty by the difference of subjective well-being among people with high social isolation

In this chapter, we focus on C4, with high social isolation, and describe the results of the Kaplan-Meier method to determine the relationship between subjective well-being and

**Table 5. Physical frailty rates by cluster.**

| cluster (n) | Physical frailty rates [#] | | | |
| --- | --- | --- | --- | --- |
| | (number of persons) | | | |
| | under 65 | Over 65, under 75 | over 75 | all |
| C1 | 10.0%[a] | 7.3%[a] | 18.6%[a] | 11.0%[a] |
| (n = 155) | (n = 3) | (n = 6) | (n = 8) | (n = 17) |
| C2 | 11.5%[a] | 10.8%[a] | 22.3%[a] | 13.7%[a] |
| (n = 701) | (n = 21) | (n = 38) | (n = 37) | (n = 96) |
| C3 | 17.8%[a] | 16.2%[a] | 31.2%[a,b] | 20.9%[b] |
| (n = 851) | (n = 24) | (n = 75) | (n = 79) | (n = 178) |
| C4 | 38.1%[b] | 28.7%[b] | 44.9%[b] | 37.0%[c] |
| (n = 246) | (n = 24) | (n = 27) | (n = 40) | (n = 91) |
| all | 17.6% | 14.7% | 29.8% | 19.6% |
| (n = 1,953) | (n = 72) | (n = 146) | (n = 164) | (n = 382) |

[#] Groups with different symbols (a-c) indicate significant differences at the 5% level.

physical frailty in this population. We used the questionnaire items on *Shiawase* and *Ikigai* as indicators of subjective well-being. Among the C4 with high social isolation, the percentage of those with physical frailty by the level of *Shiawase* was 24.3% in the group with a high level of *Shiawase*, 38.3% in that with a normal level of *Shiawase*, and 59.3% in that with a low level of *Shiawase*. Similarly, among the C4 with high social isolation, the number percentage of those with physical frailty according to the level of *Ikigai* was 23.5% in the group with a high level of *Ikigai*, 40.6% in that with a normal level of *Ikigai*, and 60.3% in that with a low level of *Ikigai*.

Fig 2 shows the results of the Kaplan-Meier analysis of the rate of physical frailty in C4 with high social isolation according to the level of *Shiawase*. The risk of physical frailty increased as the level of well-being decreased (Fig 2). Table 6 shows the average age of survival by the level of *Shiawase*. We found significant differences in all comparisons between the two groups: the high and low level of *Shiawase* ($p<0.01$), the high and normal level of *Shiawase* ($p<0.01$), and the normal and low level of *Shiawase* ($p<0.05$). At the age of 75, the group with a high level of *Shiawase* had a significantly lower risk of physical frailty by 47.6% than that with a low level of *Shiawase*.

Fig 3 shows the results of the Kaplan-Meier analysis of the rate of physical frailty in C4 with high social isolation by the level of *Ikigai*. The risk of physical frailty increased as the level of *Ikigai* decreased (Fig 3). Table 7 shows the average age of survival according to the level of *Ikigai*. We found significant differences between the group with the high and low level of *Ikigai* ($p<0.01$) and between the group with the normal and low level of *Ikigai* ($p<0.01$). At age 75, the group with a high level of *Ikigai* had a significantly lower risk of physical frailty (42.7%) than that with a low level of *Ikigai*.

## Discussion

### Relationship between social isolation and physical frailty

Participants with a higher degree of social isolation had a significantly higher rate of physical frailty than those with a lower degree of social isolation. This result was similar by age group ($< 65$ y., 65–74 y., $> = 75$ y.). The results of the present study are similar to those of several previous studies and support the hypothesis that social isolation increases the risk of physical frailty [6].

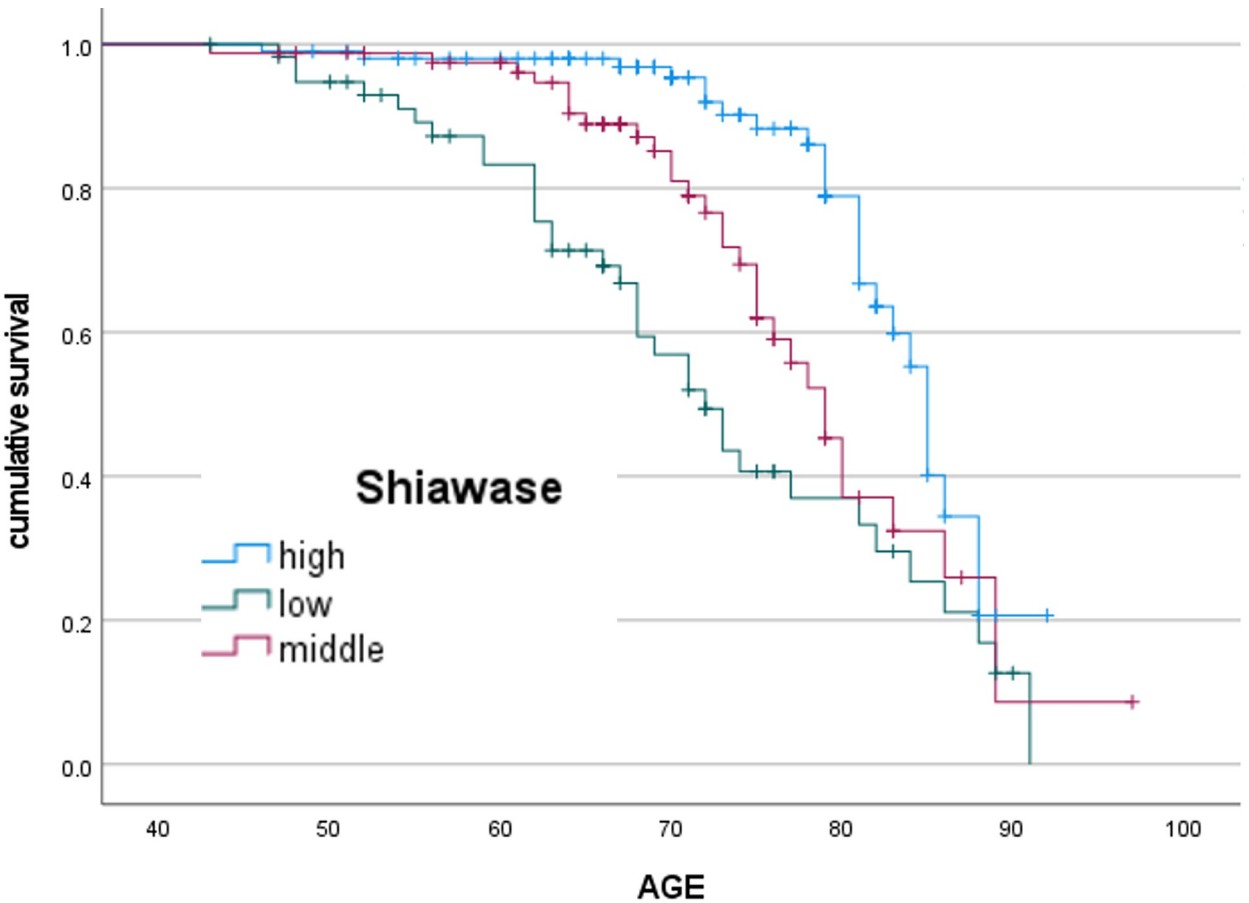

**Fig 2. Kaplan-Meier analysis according to degree of *Shiawase*.**

### Effects of *Shiawase* and *Ikigai* on physical frailty in people with high degree of social isolation

*Shiawase* and *Ikigai* in middle-aged and elderly people may affect the rate of physical frailty [21]. In the present study, we examined whether *Shiawase* and *Ikigai* influenced the rate of physical frailty in people with high social isolation. The results showed that *Shiawase* and *Ikigai* were factors that differentially affected the rate of physical frailty in people with high social isolation. In other words, even if the degree of social isolation is high, those who live with *Shiawase* or *Ikigai* may have a lower risk of physical frailty.

Although few previous studies have examined the relationship between *Shiawase* and *Ikigai* and physical frailty, several studies have reported that these factors explain low rates of frailty

**Table 6. The average age of survival by the level of *Shiawase*.**

| The level of *Shiawase* | Estimated value | Standard error | 95% confidence interval | |
|---|---|---|---|---|
| | | | Lower limit | Upper limit |
| high | 83.643 | 1.069 | 81.547 | 85.739 |
| middle | 78.939 | 1.603 | 75.796 | 82.081 |
| low | 72.964 | 1.922 | 69.198 | 76.730 |
| all | 79.977 | 0.964 | 78.087 | 81.867 |

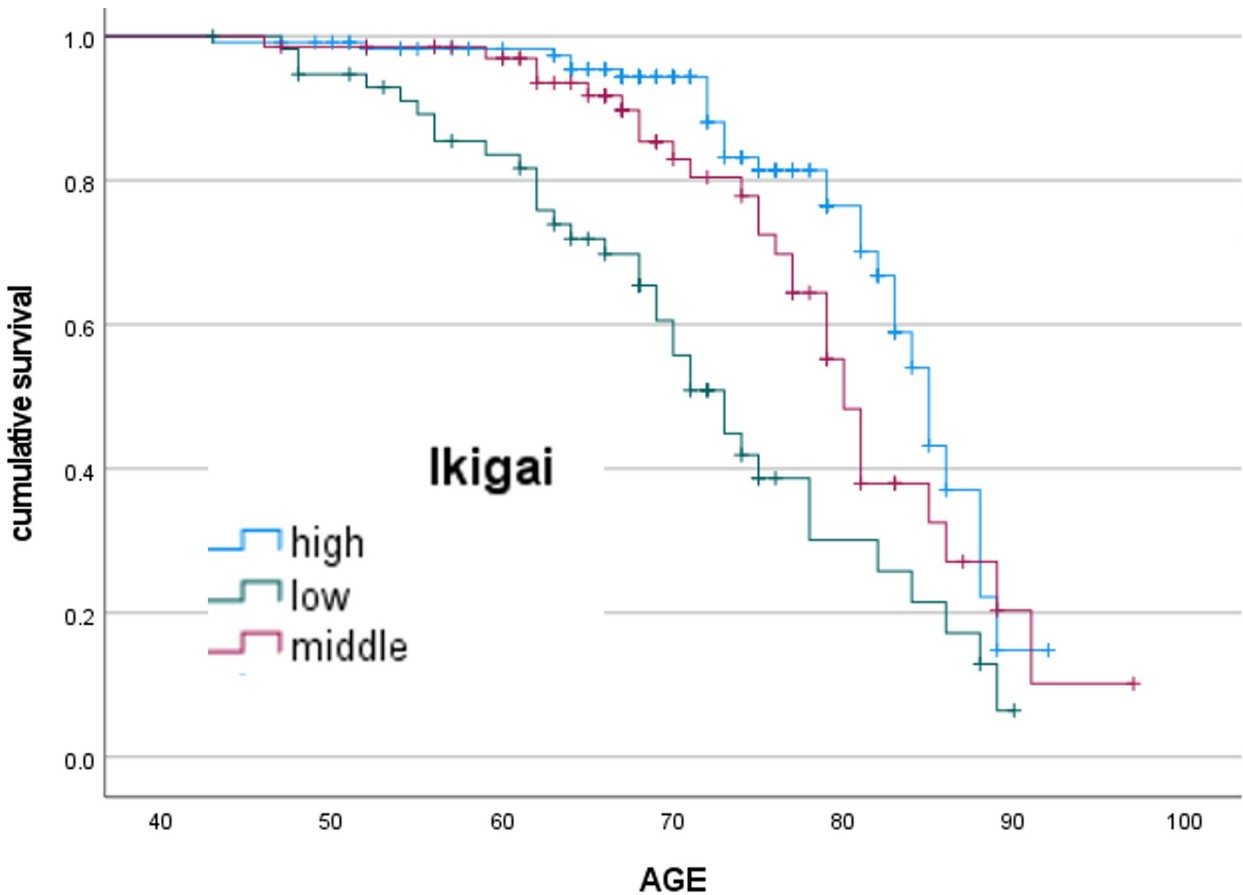

**Fig 3. Kaplan-Meier analysis according to degree of *Ikigai*.**

[21]. The present study is the first to show that *Shiawase* and *Ikigai* affect the rate of physical frailty in people with high social isolation.

### Why do people with *Shiawase* and *Ikigai* have lower rates of physical frailty even among those with high levels of social isolation?

Relatively strong evidence suggests that social isolation increases the risk of physical frailty [6]. However, in the present study, even among those with high social isolation, those with high levels of *Shiawase* and *Ikigai* maintained a low risk of physical frailty. Intentional or self-imposed social isolation might have contributed to these results.

Social isolation among older adults is diverse and includes both accidental and intentional isolation. Intentional social isolation is believed to lead to a reduction in social networks and

**Table 7. The average age of survival by the level of *Ikigai*.**

| The level of *Ikigai* | Estimated value | Standard error | 95% confidence interval | |
|---|---|---|---|---|
| | | | Lower limit | Upper limit |
| high | 82.960 | 1.080 | 80.842 | 85.077 |
| middle | 80.409 | 1.644 | 77.187 | 83.631 |
| low | 72.368 | 1.788 | 68.862 | 75.873 |
| all | 80.051 | 0.961 | 78.167 | 81.934 |

personal social interactions as a result of prioritizing emotional satisfaction and spending more time with those who are closer to them in old age [40, 41]. Tanaka et al. (2020) [41] reported that those who were socially isolated were characterized by less interaction with the community and poor physical function, although they had no history of current or prior illness. They were also highly educated, did not tend to suffer economically, and were in situations where they could deliberately choose to live in social isolation.

Toyoshima and Kusumi (2022) [42] examined the relationship between preference for solitude (which in this study refers to time spent alone) and subjective well-being and reported that enjoyment of solitude and productivity while being solitary were associated with the maintenance of subjective well-being among the elderly. In our study, the number of individuals with high levels of social isolation who were intentionally socially isolated or enjoyed solitude was unclear; however, it could be inferred that there was a certain number of them. Because among the participants with high social isolation, the percentages of those with high *Shiawase* and *Ikigai* scores were 42.4% (n = 103) and 48.4% (n = 119), respectively. These people may be intentionally socially isolated or may enjoy solitude. Even if they are socially isolated, those who are aware of their social isolation and lead a lifestyle that facilitates happiness or live with joy and the goal of living may be less likely to fall into the frailty cycle proposed by Fried et al. (2001) [23]. Although only a few studies have tested this hypothesis, the results of the present study support it.

Previous studies have shown that physical activity reduces the risk of frailty the most among physical, cultural, and community [43]. Nagai et al. (2018) [44] also found that if 30 minutes of sedentary activity time could be replaced with low-and even medium-to-high-intensity physical activity, frailty status could be improved by 16% and 42%, respectively. Furthermore, sedentary activity styles centered on TV viewing were associated with lower physical function than other sedentary activity styles (less sedentary activity, work and computer use-centered, and leisure activities other than TV viewing) [45]. Increasing the amount of physical activity performed alone and reducing the amount of time spent on passive sedentary activities may reduce the risk of physical frailty even if the degree of social isolation is high. This study did not examine the extent to which the participants preferred to be alone. In the future, it is necessary to investigate the extent to which people prefer to spend time alone as well as their social isolation status, and to investigate what kind of activities make them feel *Shiawase* or *Ikigai*.

Finally, the promotion of physical frailty prevention in middle- and older-aged adults with high levels of social isolation is discussed. A survey of the Japanese elderly population revealed that awareness of the term "frailty" was low, at about 20%, and that the more people needed to take measures against frailty, the lower their level of awareness [46]. Understanding and preventing physical frailty are important for maintaining and improving the quality of life of individuals, regardless of whether they are socially isolated or prefer to spend time alone. Therefore, it is necessary to provide information and preventive programs for frailty among older adults in social isolation. Delivering such information to socially isolated people and encouraging them to take preventive actions are subjects for future research.

## Limitations

This study had several limitations. The study design was cross-sectional; therefore, causal relationships between the variables could not be identified. In addition, all questionnaire responses were self-reported. This may have generated misclassification bias. To eliminate the influence of COVID-19 as much as possible, the participants were asked to recall two time points: 2019 (before the spread of COVID-19) and 2011. This may have resulted in inaccurate results. In

**Table 8. The difference of the analyzed and excluded data.**

| | n | Percentage of sex (Male)[#] | Percentage of age [#] | | | Average age (years) (standard deviation) | Physical frailty rates [#] |
|---|---|---|---|---|---|---|---|
| | | | under 65 | over 65, under 75 | over 75 | all | |
| analyzed data | 1,953 | 51.0%[a] | 21.0%[a] | 50.8%[a] | 28.2%[a] | 69.8 (8.42) | 19.6% [a] |
| excluded data | 1,753 | 42.7%[b] | 9.6%[b] | 34.7%b[b] | 55.7%[b] | 74.7 (8.34) | 29.4% [b] |

[#] Groups with different symbols (a, b) indicate significant differences at the 5% level.

addition, missing data were removed from 3, 706 data points collected in this study, resulting in a final sample of 1,953 participants. Because of the removal of missing data, only those who responded to all items used in the analysis were extracted. In addition, information on individuals who had never been certified as needing long-term care between 2011 and 2019 was extracted. The sociodemographic characteristics and health statuses of the included and excluded participants are shown in Table 8. In the analyzed data (n = 1,953) and the excluded data (n = 1,753), the percentage of males differed by 8.3 percentage point, the average age was 4.9 years, and the rate of physical frailty was 9.8 percentage point. Among the excluded data, physical frailty was assessed in 988 samples. These differences suggest that this study analyzed a relatively healthy sample of the total data. The survey items did not include feelings of loneliness or a preference for solitude. Investigating these items would have allowed for more multidimensional considerations. Finally, the study area is a small city in the mountainous region of Japan. Therefore, similar results may not be obtained for small, urban, or large cities.

This study has several limitations. The study design was cross-sectional, so causal relationships between variables could not be identified. In addition, all responses to the questionnaire were self-reported. This may have generated a misclassification bias. Furthermore, to eliminate the influence of COVID-19 as much as possible, the participants were asked to recall two time points: 2019 (before the spread of COVID-19) and 2011. This may have resulted in inaccurate responses. In addition, missing data were removed from the 3,706 data collected in this study, resulting in a final sample of 1,953 analyzed. As a result of the removal of missing data, only those who responded to all items used in the analysis were extracted. In addition, people who have never been certified as needing long-term care from 2011 to 2019 were extracted. The sociodemographic characteristics and health status of the analyzed and excluded data are shown in Table 8. In the analyzed data (n = 1,953) and the excluded data (n = 1,753), the percentage of males differs by 8.3 percentage point, the average age by 4.9 years, and the rate of physical frailty by 9.8 percentage point. Of the excluded data, physical frailty was assessable in 988 samples. These differences suggest that this study is an analysis of a relatively healthy sample of the total data. And the survey items did not include feelings of loneliness and preference for solitude. Investigation of these items would have allowed for more multidimensional consideration. Finally, the study area was a small city in a mountainous region of Japan. Therefore, the same results may not be obtained in other small cities, urban areas, or large cities.

## Conclusions

The research question for this study was," Does subjective well-being affect physical frailty in middle-aged and elderly Japanese with high social isolation?." The results showed that subjective well-being affected physical frailty in middle-aged and elderly Japanese with high social isolation.

In conclusion, our results suggest that subjective well-being, such as *Shiawase* and *Ikigai*, may influence physical frailty risk among middle-aged and elderly Japanese individuals with high levels of social isolation living in mountainous areas. The strength of this study is that it

suggests that living with a feeling of happiness and purpose may reduce the risk of physical frailty, even among individuals with high levels of social isolation. This study provides important insights for the prevention of frailty among socially isolated individuals. On the other hand, a shortcoming of this study is that the reasons why socially isolated individuals are happy and what kind of purpose they have in life were not investigated. These aspects require further investigation. The results of this study can serve as a reference for other countries with an aging population. The participants in this study live in a typical regional city with an aging and declining population; the average income of its citizens is about $18,000 (converted to 147.78 yen/dollar), which is relatively low among all Japanese cities [47]. In the future, many countries will face aging populations. In these countries, it is expected that the aging population, social isolation, and frail population will increase in rural areas compared with urban areas, resulting in widening regional disparities. The findings of this study suggest that physical frailty must be prevented in rural cities with aging populations, considering both the diversity of social isolation and subjective well-being of individuals.

## Supporting information

**S1 Fig. AIC scores in clustering of social isolation by PLSA.**
(TIF)

**S2 Fig. Bayesian network about C2.**
(TIF)

**S3 Fig. Bayesian network about C3.**
(TIF)

**S4 Fig. Bayesian network about C4.**
(TIF)

**S1 Table. Characteristics of each cluster by probabilistic inference.**
(TIF)

## Author Contributions

**Conceptualization:** Kai Tanabe, Yuki Sugawara, Yukihiko Okada.

**Data curation:** Yuki Sugawara, Eiichi Sakurai.

**Formal analysis:** Yuki Sugawara, Eiichi Sakurai, Yoichi Motomura, Yukihiko Okada.

**Funding acquisition:** Shinya Kuno.

**Investigation:** Kai Tanabe, Akiko Tsukao.

**Methodology:** Yuki Sugawara, Eiichi Sakurai, Yoichi Motomura, Yukihiko Okada, Akiko Tsukao.

**Project administration:** Kai Tanabe.

**Resources:** Shinya Kuno.

**Supervision:** Kai Tanabe, Shinya Kuno.

**Validation:** Yuki Sugawara, Eiichi Sakurai, Yoichi Motomura, Yukihiko Okada.

**Writing – original draft:** Kai Tanabe, Yuki Sugawara.

**Writing – review & editing:** Kai Tanabe, Yuki Sugawara, Eiichi Sakurai, Yoichi Motomura, Yukihiko Okada, Akiko Tsukao, Shinya Kuno.

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
