## [Decision Letter · Decision Letter 0]

10 Aug 2023

PONE-D-23-19593Are people with high social isolation at high risk of physical frailty? An analysis of this relationship and the impact of subjective well-beingPLOS ONE

Dear Dr. TANABE,

Thank you for submitting your manuscript to PLOS ONE. After careful consideration, we feel that it has merit but does not fully meet PLOS ONE’s publication criteria as it currently stands. Therefore, we invite you to submit a revised version of the manuscript that addresses the points raised during the review process.

We have completed the review of your manuscript and a summary is appended below. The reviewer(s) have recommended some major revisions to your manuscript.  Therefore, I invite you to respond to the reviewer(s)' comments and revise your manuscript.

We look forward to receiving your revised manuscript.

Kind regards,

Dr Sayani Das, PhD

Academic Editor

PLOS ONE

“This work was supported by a Health Labor Sciences Research Grant, Research for Building Evidence on the Results of Verification Programs for Prevention and Health Promotion (22FB1002).”

“This work was supported by a Health Labor Sciences Research Grant, Research for Building Evidence on the Results of Verification Programs for Prevention and Health Promotion (22FB1002).”

“This work was supported by a Health Labor Sciences Research Grant, Research for Building Evidence on the Results of Verification Programs for Prevention and Health Promotion (22FB1002).”

Reviewers' comments:

Reviewer's Responses to Questions

**Comments to the Author**

1. Is the manuscript technically sound, and do the data support the conclusions?

Reviewer #1: Yes

Reviewer #2: Yes

2. Has the statistical analysis been performed appropriately and rigorously? 

Reviewer #1: Yes

Reviewer #2: Yes

3. Have the authors made all data underlying the findings in their manuscript fully available?

Reviewer #1: Yes

Reviewer #2: No

4. Is the manuscript presented in an intelligible fashion and written in standard English?

Reviewer #1: Yes

Reviewer #2: Yes

5. Review Comments to the Author

Reviewer #1: The article identifies a relevant knowledge gap that the authors attempt to address, however, in this context, it would be important to also provide a brief description for the motivation of this study. There have been previous studies that have also associated physical frailty among older adults with a range of other health outcomes such as falls, body composition, obesity, ADL etc. Thus, a justification seems to be pertinent in this case where the authors clarify their motivation for studying social isolation with physical frailty. Adding a paragraph explicating what has been studied so far with respect to physical frailty could be beneficial focusing on the “gap” in the literature that this study aims to close. More work needs to be done in the conclusion section. As it currently stands, the conclusion is rather limited and there is more scope to build upon the implications of the findings obtained from the study. Conclusion should be elaborated upon articulating what are keys takeaways from the article. The areas that need further strengthening are: strengths and weaknesses, implications for practice, and conclusion. The information around implication is insufficient and generates a superficial understanding. The article should be proofread by a native English speaker. Multiple grammatical errors and awkward phrasing were found throughout the paper.

Regarding the contribution that is interesting for an international audience (or is it too focus on a specific national context)?: At present as the article stands, it would require more holistic understanding of how the notion of frailty is perceived within different country contexts, such as, lower-and-middle income countries (LMICs) as opposed to developed countries. The article should include studies from different study contexts to attract a larger number of readers. Currently, the article focuses particularly on Japan and the Japanese notion of well-being which, if constructively utilized, could be very well situated within the broader domain of aging and geriatrics to compare different studies. This would, in fact, also attract international readers and will present a higher chance of citation.

Additional comments: While the abstract is well written, however, explicit statements about the contributions of the article is largely absent. Further, the abstract does touch upon the knowledge gap but not sufficiently. The authors could try to add a few more sentences to present the gap in the literature and how the present article is particularly attempting to close that gap with the research findings.

Reviewer #2: This study investigates the correlation between social isolation, subjective well-being, and physical frailty in middle-aged and older Japanese adults residing in a mountainous area. The research holds great importance in advancing our knowledge of strategies for promoting healthy aging. Furthermore, the manuscript is well-structured and thoughtfully written, providing readers with a clear comprehension of the research's motivations, study design, analyses, and results. However, in my opinion, there are a few issues that, if addressed, could further enhance the manuscript's quality.

• I recommend that the author consider revising both the title and the primary aim of this study (in the introduction section) in order to better align them with the research question, conducted analyses, and the focus of the discussion section, which centers around "the relationship between subjective well-being and frailty in the high social isolation group."

• Within the method section, the author has indicated the existence of two data sources. To enhance clarity, please specify the variables obtained from each respective source.

• The author mentioned that “to eliminate the influence of COVID-19 as much as possible, the participants were asked to recall two time points: 2019 (before the spread of COVID-19) and 2011”. Could you kindly provide clarification on the specific questions or variables for which the participants were asked to recall their conditions in 2019 and 2011?

• According to the data provided in the participants section, approximately 51% (2,092 out of 4,045) of the participants were excluded from the analysis. The author has acknowledged in the limitation section that this exclusion "may have biased the analysis." Could the author provide further elaboration on the implications of this bias and its potential impact on the validity and credibility of this study?

• Furthermore, it would be better if the author could elucidate the distinctions between the excluded sample and the analytical sample, specifically concerning their sociodemographic characteristics and health status.

• In my opinion, the final paragraph of the discussion section lacks coherence with the preceding paragraph. I would recommend revising it to ensure a smoother transition between the two paragraphs.

• In the introduction section, the research question was formulated as "Are all people with a high degree of social isolation at higher risk for physical frailty?" whereas, in the conclusions section, it was rephrased as "Are people with a high degree of social isolation generally at higher risk for physical frailty?" Although the two sentences share similarities, I recommend revising the one in the conclusions section to avoid any confusion.

6. PLOS authors have the option to publish the peer review history of their article (what does this mean?). If published, this will include your full peer review and any attached files.

Reviewer #1: No

Reviewer #2: No

---

## [Author Response · Author response to Decision Letter 0]

10 Oct 2023

Reviewer1: We wish to express our appreciation to the reviewers for their insightful comments on our paper. The comments have helped us significantly improve the paper.

Reviewer2: We wish to express our strong appreciation to the reviewers for their insightful comments on our paper. We feel the comments have helped us significantly improve the paper.

---

## [Decision Letter · Decision Letter 1]

7 Nov 2023

PONE-D-23-19593R1Impact of subjective well-being on physical frailty in middle-aged and elderly Japanese with high social isolationPLOS ONE

Dear Dr. TANABE,

Thank you for submitting your manuscript to PLOS ONE. After careful consideration, we feel that it has merit but does not fully meet PLOS ONE’s publication criteria as it currently stands. Therefore, we invite you to submit a revised version of the manuscript that addresses the points raised during the review process.

We have completed the review of your manuscript and a summary is appended below. The reviewer(s) have recommended some minor revisions to your manuscript.  Therefore, I invite you to respond to the reviewer(s)' comments and revise your manuscript.

We look forward to receiving your revised manuscript.

Kind regards,

Sayani Das, PhD

Academic Editor

PLOS ONE

Journal Requirements:

Reviewers' comments:

Reviewer's Responses to Questions

**Comments to the Author**

1. If the authors have adequately addressed your comments raised in a previous round of review and you feel that this manuscript is now acceptable for publication, you may indicate that here to bypass the “Comments to the Author” section, enter your conflict of interest statement in the “Confidential to Editor” section, and submit your "Accept" recommendation.

Reviewer #1: (No Response)

Reviewer #2: All comments have been addressed

2. Is the manuscript technically sound, and do the data support the conclusions?

Reviewer #1: Yes

Reviewer #2: Yes

3. Has the statistical analysis been performed appropriately and rigorously? 

Reviewer #1: Yes

Reviewer #2: Yes

4. Have the authors made all data underlying the findings in their manuscript fully available?

Reviewer #1: No

Reviewer #2: Yes

5. Is the manuscript presented in an intelligible fashion and written in standard English?

Reviewer #1: Yes

Reviewer #2: Yes

6. Review Comments to the Author

Reviewer #1: While the authors have made necessary revisions to improve the article, there still remains a few areas that could be reworked upon.

1. The author(s) should also provide a descriptive statistics table with mean and standard deviation (SD) of age, Ikigai scores, and Shiawase scores, frailty, and social isolation scores (essentially the scores that have been measured for this study).

2. The RQ “Are all people with a high degree of social isolation at higher risk for physical frailty?” remains too broad. It still misses the ‘subjective well-being’ component which is as important part of the study. Additionally, the demographics (middle aged and elderly Japanese) could be added in the RQ as well. “All people” does not stand correct in the current phrasing and is too broad.

3. The author(s) in line 78-80 state(s) that “for physical frailty, there is a need to consider the diversity of social isolation in the prevention of physical frailty. However, no studies have yet been conducted to clarify this issue”. Considering the diversity of social participation on physical frailty, the authors could refer to the following work in the context of China:

Xie, B., & Ma, C. (2021). Effect of social participation on the development of physical frailty: Do type, frequency and diversity matter? Maturitas, 151, 48-54.

4. To make the study findings stronger, is it possible to supply the employment/occupation information of the participants in the descriptive statistics table? The author(s) report the study of Tanaka et al. (2020) in the context of Japan that state those who were socially isolated were highly educated, weren’t financially disadvantaged and chose to deliberately live in social isolation. However, in this study, the authors state that the participants of the city had an average income of $18000 which is not a lot, but we still do not know the employment/income status of the participants in this study. Including income and education as part of the descriptive statistics table is highly recommended. The correlation (if not causation) between income/employment and social isolation could be really interesting given the data is cross-sectional. You could maybe add some sentences of how this study findings are consistent with Tanaka et al. or not in the discussion section.

5. Although the authors do note that the purpose of social isolation has not been explored, but perhaps providing their employment/income status could be a good reference point for future studies. It could help future researchers identify whether income/education really matter when it comes to individuals isolating themselves in other country contexts.

6. On lines 341 the author(s) mention that “In our study, the number of individuals with high levels of social isolation who were intentionally socially isolated or enjoyed solitude was unclear; however, it could be inferred that there was a certain number”. Can you explicitly state the number? Or what exactly is it that you are referring to (how do you infer)? At present, it is confusing and unclear to the reader.

Reviewer #2: I thank the authors for thoroughly addressing my comments. The manuscript has undergone significant improvements. As of now, I have no further comments to make.

7. PLOS authors have the option to publish the peer review history of their article (what does this mean?). If published, this will include your full peer review and any attached files.

Reviewer #1: No

Reviewer #2: No

---

## [Author Response · Author response to Decision Letter 1]

21 Dec 2023

RESPONSE TO REVIEWER #1:

Reviewer #1 comment 1: 

1. The author(s) should also provide a descriptive statistics table with mean and standard deviation (SD) of age, Ikigai scores, and Shiawase scores, frailty, and social isolation scores (essentially the scores that have been measured for this study).

Answer）

Thank you for your suggestion. Table 3 shows that information. In addition, variable scores were appended to Table 2.

Additions & Corrections）

Section: Results

> Table 3 and P17 L236-246

The averages and standard deviations for age, Shiawase score, Ikigai score, physical frailty score, and social isolation score are shown in Table 3. Social isolation scores in this study were calculated by summing the scores of the social isolation questionnaire items (Table 2), in which 1 point was allocated for each of the following: living alone; not having a spouse; frequency of conversation with family members is not daily; not participating in any of the social activities, respectively; not eating with someone once a day; frequency of going out decreasing; not employed. In addition, the frequency of conversation with non-family and the frequency of conversation with phone or SNS, community cooperating trust, willing of contribution to community was divided into three-point scales: high, normal, and low, and assigned a score of 1, 2, and 3, respectively. In addition, the number and percentage of people in the analyzed data for employment status and education status are shown in Table 3.

Section: Methods

> Table 2 

The scores of the variables were added in the table.

Reviewer #1 comment 2: 

The RQ “Are all people with a high degree of social isolation at higher risk for physical frailty?” remains too broad. It still misses the ‘subjective well-being’ component which is as important part of the study. Additionally, the demographics (middle aged and elderly Japanese) could be added in the RQ as well. “All people” does not stand correct in the current phrasing and is too broad.

Answer）

We agree with you. We have revised the following based on your suggestion.

Additions & Corrections)

Section: Introduction

> P6 L90-91

Section: Conclusions

> P30 L423-424

Before revision) Are all people with a high degree of social isolation at higher risk for physical frailty?

After revision) Does subjective well-being affect physical frailty in middle-aged and elderly Japanese with high social isolation?

Reviewer #1 comment 3:

The author(s) in line 78-80 state(s) that “for physical frailty, there is a need to consider the diversity of social isolation in the prevention of physical frailty. However, no studies have yet been conducted to clarify this issue”. Considering the diversity of social participation on physical frailty, the authors could refer to the following work in the context of China:

Xie, B., & Ma, C. (2021). Effect of social participation on the development of physical frailty: Do type, frequency and diversity matter? Maturitas, 151, 48-54.

Answer）

Thank you for providing these insights. We agree with you and have incorporated the suggestion into the introduction of our paper.

Additions & Corrections)

Section: Introduction

> P5 L78-80

Before revision) Similarly for physical frailty, there is a need to consider the diversity of social isolation in the prevention of physical frailty. However, no studies have yet been conducted to clarify this issue.

After revision) Similarly for physical frailty, there is a need to consider the diversity of social isolation in the prevention of physical frailty. However, only a few studies have clarified this issue [14].

[14] Xie B, Ma C. Effect of social participation on the development of physical frailty: Do type, frequency and diversity matter? Maturitas. 2021;151: 48-54. doi: https://doi.org/10.1016/j.maturitas.2021.06.015

Reviewer #1 comment 4~6: 

4. To make the study findings stronger, is it possible to supply the employment/occupation information of the participants in the descriptive statistics table? The author(s) report the study of Tanaka et al. (2020) in the context of Japan that state those who were socially isolated were highly educated, weren’t financially disadvantaged and chose to deliberately live in social isolation. However, in this study, the authors state that the participants of the city had an average income of $18000 which is not a lot, but we still do not know the employment/income status of the participants in this study. Including income and education as part of the descriptive statistics table is highly recommended. The correlation (if not causation) between income/employment and social isolation could be really interesting given the data is cross-sectional. You could maybe add some sentences of how this study findings are consistent with Tanaka et al. or not in the discussion section.

5. Although the authors do note that the purpose of social isolation has not been explored, but perhaps providing their employment/income status could be a good reference point for future studies. It could help future researchers identify whether income/education really matter when it comes to individuals isolating themselves in other country contexts.

6. On lines 341 the author(s) mention that “In our study, the number of individuals with high levels of social isolation who were intentionally socially isolated or enjoyed solitude was unclear; however, it could be inferred that there was a certain number”. Can you explicitly state the number? Or what exactly is it that you are referring to (how do you infer)? At present, it is confusing and unclear to the reader.

Answer）

We thank the reviewer for these comments. Comments 4 through 6 were similar and were therefore answered together in this section.

In our study, 7.3% of participants with high levels of social isolation were highly educated (college graduates or higher). However, we avoided discussing this study because we lacked data (income and economic status) to examine its consistency with the study by Tanaka et al. (2020).

Additions & Corrections)

Section: Results

> Table 3 and P17 L236-246

The number and percentage of people in the analyzed data for employment status and education status are shown in Table 3.

(Income data was not obtained and could not be displayed.)

Section: Discussion

> P26 L350-355

Before revision) In our study, the number of individuals with high levels of social isolation who were intentionally socially isolated or enjoyed solitude was unclear; however, it could be inferred that there was a certain number.

After revision) In our study, the number of individuals with high levels of social isolation who were intentionally socially isolated or enjoyed solitude was unclear; however, it could be inferred that there was a certain number of them. Because among the participants with high social isolation, the percentages of those with high Shiawase and Ikigai scores were 42.4% (n=103) and 48.4% (n=119), respectively. These people may be intentionally socially isolated or may enjoy solitude.

---

## [Decision Letter · Decision Letter 2]

14 Jan 2024

Impact of subjective well-being on physical frailty in middle-aged and elderly Japanese with high social isolation

PONE-D-23-19593R2

Dear Dr. TANABE,

We’re pleased to inform you that your manuscript has been judged scientifically suitable for publication and will be formally accepted for publication once it meets all outstanding technical requirements.

Kind regards,

Sayani Das, PhD

Academic Editor

PLOS ONE

Additional Editor Comments (optional):

Reviewers' comments:

Reviewer's Responses to Questions

**Comments to the Author**

1. If the authors have adequately addressed your comments raised in a previous round of review and you feel that this manuscript is now acceptable for publication, you may indicate that here to bypass the “Comments to the Author” section, enter your conflict of interest statement in the “Confidential to Editor” section, and submit your "Accept" recommendation.

Reviewer #1: All comments have been addressed

Reviewer #2: All comments have been addressed

2. Is the manuscript technically sound, and do the data support the conclusions?

Reviewer #1: Yes

Reviewer #2: Yes

3. Has the statistical analysis been performed appropriately and rigorously? 

Reviewer #1: Yes

Reviewer #2: Yes

4. Have the authors made all data underlying the findings in their manuscript fully available?

Reviewer #1: No

Reviewer #2: No

5. Is the manuscript presented in an intelligible fashion and written in standard English?

Reviewer #1: Yes

Reviewer #2: Yes

6. Review Comments to the Author

Reviewer #1: The author(s) have addressed all concerns in the revision sufficiently. I have no further comments. Will be happy to cite this work in the future.

Reviewer #2: (No Response)

7. PLOS authors have the option to publish the peer review history of their article (what does this mean?). If published, this will include your full peer review and any attached files.

Reviewer #1: No

Reviewer #2: No

---

## [Editor Report · Acceptance letter]

17 Feb 2024

PONE-D-23-19593R2 

PLOS ONE

Dear Dr. TANABE, 

I'm pleased to inform you that your manuscript has been deemed suitable for publication in PLOS ONE. Congratulations! Your manuscript is now being handed over to our production team.

Kind regards, 

on behalf of

Dr Sayani Das 

Academic Editor

PLOS ONE